# Mycobiota of Mexican Maize Landraces with Auxin-Producing Yeasts That Improve Plant Growth and Root Development

**DOI:** 10.3390/plants12061328

**Published:** 2023-03-15

**Authors:** Juan Ramos-Garza, José Luis Aguirre-Noyola, Rafael Bustamante-Brito, Lily X. Zelaya-Molina, Jessica Maldonado-Hernández, Aurea Itzel Morales-Estrada, Zoe Resendiz-Venado, Jacqueline Palacios-Olvera, Thania Angeles-Gallegos, Paola Terreros-Moysen, Manuel Cortés-Carvajal, Esperanza Martínez-Romero

**Affiliations:** 1Escuela de Ciencias de la Salud, Campus Coyoacán, Universidad del Valle de México, Calzada de Tlalpan 3016/3058, Coapa, Ex Hacienda Coapa, Coyoacán 04910, Ciudad de México, Mexico; 2Programa de Ecología Genómica, Centro de Ciencias Genómicas, Universidad Nacional Autónoma de México, Av. Universidad s/n, Cuernavaca 62210, Morelos, Mexico; 3Laboratorio de Recursos Genéticos Microbianos, Centro Nacional de Recursos Genéticos-INIFAP, Boulevard de la Biodiversidad No. 400, Tepatitlán de Morelos 47600, Jalisco, Mexico

**Keywords:** biofertilizers, bioinoculants, corn, plant microbiota, root exudates, sustainable agriculture, *Zea mays*

## Abstract

Compared to agrochemicals, bioinoculants based on plant microbiomes are a sustainable option for increasing crop yields and soil fertility. From the Mexican maize landrace “Raza cónico” (red and blue varieties), we identified yeasts and evaluated in vitro their ability to promote plant growth. Auxin production was detected from yeast isolates and confirmed using *Arabidopsis thaliana* plants. Inoculation tests were performed on maize, and morphological parameters were measured. Eighty-seven yeast strains were obtained (50 from blue corn and 37 from red corn). These were associated with three families of Ascomycota (*Dothideaceae*, *Debaryomycetaceae*, and *Metschnikowiaceae*) and five families of Basidiomycota (*Sporidiobolaceae, Filobasidiaceae, Piskurozymaceae*, *Tremellaceae*, and *Rhynchogastremataceae*), and, in turn, distributed in 10 genera (*Clavispora*, *Rhodotorula*, *Papiliotrema*, *Candida*, *Suhomyces*, *Soliccocozyma*, *Saitozyma Holtermaniella*, *Naganishia,* and *Aeurobasidium*). We identified strains that solubilized phosphate and produced siderophores, proteases, pectinases, and cellulases but did not produce amylases. *Solicoccozyma* sp. RY31, *C. lusitaniae* Y11, *R. glutinis* Y23, and *Naganishia* sp. Y52 produced auxins from L-Trp (11.9–52 µg/mL) and root exudates (1.3–22.5 µg/mL). Furthermore, they stimulated the root development of *A. thaliana*. Inoculation of auxin-producing yeasts caused a 1.5-fold increase in maize plant height, fresh weight, and root length compared to uninoculated controls. Overall, maize landraces harbor plant growth-promoting yeasts and have the potential for use as agricultural biofertilizers.

## 1. Introduction

Since the second industrial revolution, poor management, excessive use, and mistimed application of chemical fertilizers and pesticides in agriculture have had negative effects on public health and the environment. High concentrations of agrochemicals cause acidification, physical deterioration, a reduction in soil biodiversity, and an increase in greenhouse gas emissions [1]. In water, the main effects of fertilizers are eutrophication, leaching, and the contamination of aquifers [2].

Corn (*Zea mays*) has become the leading cereal grain by global production volume, with around 197 million hectares of land cultivated annually [3]. In Mexico, maize’s center of origin and domestication, some landraces have adapted to different environmental conditions and are locally grown [4]. Maize landraces are reservoirs of genetic diversity and have been used as donor material for global breeding activities. The production and commercialization of these varieties could be enhanced using biofertilizers made from microbes that are natural maize symbionts. Recently, we described plant growth-promoting filamentous fungi (PGPF) associated with Mexican maize landraces. We reported strains belonging to *Penicillium*, *Didymella*, *Fusarium*, *Aspergillus*, *Clonostachys*, *Microsphaeropsis*, *Cladosporium*, *Trichoderma*, *Talaromyces*, and *Epicoccum*. Some strains showed antagonistic properties against *Fusarium* complexes that damage maize crops [5].

Bioinoculants based on microorganisms that promote plant growth and naturally establish symbiosis with crops in the field can be used to achieve sustainable agriculture [6]. They can be applied to the soil, seed, or leaf zone to allow microbes to inhabit the rhizosphere and improve nutrient availability or internally colonize plant tissues and provide beneficial effects [7]. Most commercial biofertilizers are composed of bacteria (e.g., *Azospirillum*, *Azotobacter*, and *Bacillus*) and arbuscular mycorrhizal fungi (e.g., *Glomus* and *Gigaspora margarita*) or filamentous fungi (e.g., *Trichoderma*) and their positive impacts on crop yields have been widely reported [8,9]. The use of yeasts has been popularized in various areas, such as the fermentation and food industry, enzyme biotechnology, and more recently in agriculture, for their ability to stimulate plant growth [10].

Yeasts are polyphyletic unicellular fungi that are also members of the plant and soil microbiota; they are present in small populations compared to filamentous fungi and bacteria [11]. Most yeasts tolerate a wide range of temperatures and are considered innocuous for plants and animals. They lack plasmids, reducing the risk of pathogenicity [12]. They can produce diverse compounds that are useful to plants, such as amino acids, vitamins, polyamines, enzymes, and phytohormones [13,14]. Fermenting yeast can solubilize P or K minerals in the soil by producing large quantities of citric acid and other organic acids [15]. Conversely, other yeasts produce siderophores to capture iron and make it available to plant cells [16].

Yeast can also act as a biocontrol agent of pathogenic fungi due to competition for nutrients and space, volatile organic compound secretion, lytic enzyme production, and mycoparasitism [17]. Additionally, cell wall components and glycopeptides of yeasts can induce systemic responses in plants (e.g., the production of phytoalexins, alkaloids, and glucanases) and prevent infection by *Pseudomonas syringae*, and *Botrytis cinerea* [18]. Systemic responses to abiotic stresses such as drought and salinity can be orchestrated by yeasts producing the enzyme 1-aminocyclopropane-1-carboxylate deaminase (ACC-deaminase) while providing ammonium and decreasing ethylene levels in plants [13]. Although all these outstanding characteristics and metabolic capabilities of yeasts have been reported, very few yeast species have been used in bioinoculants. For example, *Aureobasidium pullulans* (Blossom Protect™, BoniProtect™, and Botector™), *Candida oleophila* (Nexy™), and *Metschnikowia fructicola* (Noli™ and Shemer™); most are antagonists against *Erwinia amylovora*, *Botrytis cinerea*, and *Colletotrichum gloeosporioides* [12]. Therefore, further studies are needed to understand how yeasts interact with crops and how they affect, in both short- and long-term studies, plant, and soil health in fields.

Auxin production is considered one of the most important factors in a proposed bioinoculant because auxins are phytohormones that stimulate cell elongation, cell division, and vascular system differentiation [19]. Furthermore, they induce root formation and are involved in tolerance to abiotic stress [20]. Yeasts belonging to the phylum Ascomycota produce higher levels of auxins, including indole-3-acetic acid (IAA), than members of the Basidiomycota phylum [21]. Indeed, strains from the rhizosphere are good auxin producers.

In this work, we described the plant growth-promoting activities of yeasts isolated from different tissues of the Mexican maize landrace “Raza cónico” (red and blue varieties). We also analyzed auxin biosynthesis by the strains through induction assays with maize root exudates and using the model plant *Arabidopsis thaliana*. Finally, the four strains with the most outstanding characteristics were used as bioinoculants on maize plants in pot trials, and morphometric parameters were evaluated.

## 2. Results and Discussion

The intensive use of synthetic fertilizers and pesticides to improve crop production has caused the deterioration of natural resources, and has caused damage to human health. Developing sustainable agricultural practices that take advantage of crop-associated microbiota to promote plant growth and soil fertility is therefore crucial. There is a need to isolate local yeast strains from native crops and evaluate their ability to stimulate plant growth and be used in biofertilizers.

### 2.1. Yeast Diversity Associated with Maize Plants

A total of 87 yeast strains were obtained from the Mexican maize landrace “Raza cónico,” of which 50 were isolated from the blue variety and 37 from the red variety (Appendix A). In blue corn, 29 strains came from the phyllosphere, and the remaining strains were found as leaf endophytes. In the red maize, there was a greater distribution of yeasts: nine strains were isolated from kernels, 12 from the phyllosphere, 15 from the root endosphere, and one from the leaf endosphere. Phylogenetic analysis placed the isolates within the two phyla Ascomycota and Basidiomycota, eight families and 10 genera (*Clavispora*, *Rhodotorula*, *Papiliotrema*, *Candida*, *Suhomyces*, *Soliccocozyma*, *Saitozyma*, *Holtermaniella*, *Naganishia*, and *Aeurobasidium*) (Figure 1).

Most of the isolates belonged to the phylum Ascomycota [blue maize (52%) and red maize (57%)]. In contrast, in a study of maize plants in Thailand, 217 yeast isolates were obtained, and those from the phylum Basidiomycota were more abundant (81.6%) [22]. In contrast, yeasts associated with corn in Brazil belonged entirely to the phylum Ascomycota [23]. These findings showed that yeast diversity appears to be largely influenced by maize variety, tissue type, and the soil in which the plants are grown. The different maize varieties had contrasting anthocyanin content and profiles of metabolites exuded by roots which probably affected the mycobiota structure in the rhizosphere and endosphere [24].

Distribution of yeast genera in this work is shown in Figure 2. Strains of *Aeurobasidium*, *Naganishia,* and *Saitozyma* were obtained only from blue corn, and *Suhomyces*, *Holtermanniella*, *Kurtzmaniella*, *Papiliotrema,* and *Solicoccozyma* were exclusive to red maize plants. In common, we found strains of *Clavispora*, *Papiliotrema*, *Rhodotorula,* and *Candida*.

Here, *Clavispora lusitaniae* was the most predominant species (Figure 2). The monophyletic genus *Clavispora* (Anamorph: *Candida*) belongs to the *Metschnikowiaceae* family, which includes isolates from plants, fruits, and soil [25,26]. Other members of this family are well-known for producing exometabolites (e.g., 3-amino-5-methylhexanoic acid) that provide protection against post-harvest pathogens. *Rhodotorula* is an excellent endophyte; it reaches up to 90% root colonization after 8 days of inoculation of plants [27], which explains why it was the second most abundant in both maize varieties. *Papiliotrema* and *Naganishia* (previously classified as *Cryptococcus*) have also been described as members of the microbiota in soil, roots, and phyllosphere [28]. These yeasts can colonize diverse environments due to their production of capsules and melanin that help them adapt to soils with low nutrients, desiccation, and UV rays [29]. Some species of the genera *Candida* and *Suhomyces* have been described as symbionts of sugarcane and maize plants [22,30], while *Aerobasidium,* which is a notable biocontrol agent, is more commonly found in fruits and vegetables [31]. On the other hand, we found a low prevalence of the genera *Solicoccozyma*, *Saitozyma,* and *Holtermmaniella,* which are well known for their ability to release auxins and lipids [32,33,34].

### 2.2. Plant Growth-Promoting Characteristics

Using a set of biochemical tests and enzyme assays, we explored the potential of yeast strains to promote plant growth.

In soil, most phosphorus is found in minerals (such as apatite and hydroxyapatite) that are not assimilated by plants. We found that 14 yeasts could solubilize tricalcium phosphate (Appendix A). Among these, the best phosphate solubilizers were: *Rhodotorula mucilaginosa* RY1 and RY3, *Clavispora lusitaniae* RY2, *Suhomyces prunicola* RY4, and *Kurtzmaniella quercitrusa* RY11. The latter three were formally described as *Candida* and display high fermentation rates. Under phosphate starvation, corn roots exude more sugars [35]. These sugars can be fermented by yeasts producing organic acids (e.g., citric acid) that dissolve inorganic phosphate, making it bioavailable in the rhizosphere [36]. Phosphate-solubilizing yeasts have increased the biomass, shoot height, and cellular inorganic phosphate content of plants growing under phosphate limitation [37]. Improved maize plant growth and phosphate nutrition have been observed in response to inoculation with a phosphate-solubilizing *Candida railenensis*, as well as synergism in phosphate uptake by arbuscular mycorrhizal fungi [38]. Therefore, these strains would be useful for agriculture in phosphorus-deficient soils.

Siderophores are low-weight molecules that can bind to ferric ion and transport it into the cell. There, it is reduced to its ferrous form, which is bioavailable. In this study, *Papiliotrema flavecens* Y27 and *Rhodotorula glutinis* Y41 produced siderophores under iron-limiting conditions, reaching an activity of 87 ± 4% and 57 ± 3% SU, respectively (Appendix A). In contrast, *Papiliotrema* strains isolated from wild cherry tomato rhizosphere did not show siderophores production, but *Rhodotorula* strains did [39]. Biosynthesis of rhodotorulic acid, a dihydroxamate siderophore, by *Rhodotorula* has been reported with antagonistic activity against *B. cinerea* [40]. This suggests that some plant yeasts use siderophores to solubilize iron in soils or as a competitive strategy against other microorganisms in the phyllosphere [41]. Siderophores can also bind other divalent or trivalent cations, such as Co, Cu, Zn, As, Cr, Pb, and Cd [42]. Siderophore-producing yeasts allow plant growth under metal stress [43]. These yeasts have been successfully applied as bioinoculants to assist phytoremediation processes in metal-contaminated soils [44].

Enzyme activities were detected in 13 isolates, of which cellulase (44%), pectinase (27%), and protease (27%) production were the most frequent (Appendix A). Some isolates were positive for more than one enzyme. *Candida* sp. RY26, *Papiliotrema flavescens* RY32, *A*. *pullulans* Y64, *P. flavecens* Y65 and *R. glutinis* Y41, Y43, Y44, and Y58 showed cellulolytic capacity. Recently, cellulases from *Rhodotorula* have been used in the food processing and biofuel industries [45]. On the other hand, pectinolytic activity was more evident in *Solicoccozyma* sp. RY30, *A. pullulans* Y64, and *Naganishia* sp. Y19, Y52 and Y53. These isolates were found as root and leaf endophytes. Yeasts may use cellulases and pectinases as colonization mechanisms in plant tissues through the partial degradation of the cell wall [46]. The production of these enzymes and others such as β-1,3-glucanase and chitinase may also serve as biocontrol mechanisms by destroying the cell wall of phytopathogenic fungi such as *Aspergillus flavus*, *B. cinerea*, *Penicillium expansum*, *Sphaerotheca fuliginea*, and *Podosphaera xanthii* [17,47]. *Aureobasidium pullulans* Y64 and *R. glutinis* Y41, Y43, Y57, and Y58 were positive for proteases. Proteases enhance an endophytic lifestyle by increasing nitrogen uptake from the apoplast during plant colonization [48]. None of the isolates produced amylases. Yeasts that did not show enzymatic activity could use alternative colonization mechanisms, including entry through injuries or cracks. Alternatively, they could colonize through vectors, such as insects or nematodes [28].

### 2.3. Auxin Production

Auxins control several functions in plants, such as vascular tissue differentiation, root initiation, cell division, and elongation. Hence it is crucial that bioinoculants include strains that produce this phytohormone. Auxin production was detected in YPD cultures of 4 out of 87 strains: *Solicoccozyma* sp. RY31, *C. lusitaniae* Y11, *R. glutinis* Y23, and *Naganishia* sp. Y52. Indolic compound concentration increased 3 three times when tryptophan was present (Table 1). The auxin biosynthetic pathways in yeasts have not yet been fully determined; however, it is known that their main products are indole-3-acetic acid (IAA) and indole-3-pyruvic acid (IPYA) from L- Tryptophan (Trp) [49,50]. This amino acid is transformed into pyruvic acid by aminotransferases and then carboxylated to produce indole-3-acetaldehyde (IAAld) by indole-3-pyruvate decarboxylase. Finally, IAAld is oxidized to IAA. Another scenario is that IPYA is transformed to IAA by indole-3-pyruvate monooxygenase in a single step in yeast [51,52]. Analyses by high-performance liquid chromatography (HPLC) or Gas Chromatography-mass spectrometry (GC-MS) need to be performed to identify the type of auxin released by each strain. The auxin concentrations observed here (from 11.9 ± 1.9 to 52 ± 4.2 µg/mL), along with those reported by Nassar et al. [50], show that yeasts isolated from corn have a higher rate of production than those associated with non-crop plants such as *Flaveria angustifolia*, *Sphaeralcea angustifolia*, and *Prosopis* sp. [29].

The strains investigated in this study were able to produce auxins from root exudates, but this was at a lower concentration than those observed in cultures with L-Trp (Table 1). Maize exudates may have L-Trp or its precursors (anthranilate or chorismate) that would support the auxin biosynthesis pathway in yeast [24]. Interestingly, the concentration of auxins produced by the yeasts was higher in the exudates of the maize variety from which they were isolated. Strains likely have a higher affinity to uptake nutrients released by their host. It has previously been reported that *Williopsis saturnus*, an endophyte yeast, synthesized IAA by obtaining precursors from maize roots [50]. IAA also plays a role as a metabolic regulator that allows fungal cells to enter a quiescent state [53], and in complex communities, this hormone restricts the growth of neighboring fungi [54]. Therefore, auxin production by yeasts could increase their competitiveness and ability to colonize the rhizosphere and plant tissues.

### 2.4. Arabidopsis Thaliana-Yeast Interaction Assays

We extended the study to confirm the plant growth-promoting effect of auxin-producing yeasts using *A. thaliana* plants (Figure 3A). We observed that *Solicoccozyma* sp. RY31, *C. lusitaniae* Y11, and *Naganishia* sp. Y52 caused a reduction in primary root elongation as there was an increase in the number of lateral roots compared to the non-inoculated control (Figure 3B,C). We observed the formation of root hairs on seedlings inoculated with yeast (3A). It is likely that the yeasts induced the cell cycle of the root stele, and consequently, more lateral roots were formed [37]. This modification of the root structure has been stimulated by other yeasts (e.g., *Hannaella coprosmaensis*, *Ustilago esculenta*, *Sporidiobolus ruineniae*, *Pseudozyma aphidis*, and *Dothideomycetes* sp.) in *Nicotiana benthamiana* and the effect have been linked to auxin production in a concentration-dependent manner [55,56]. In our assay, the growth-promoting activity was most prominent during the interaction with the root endophyte *Solicoccozyma* sp. RY31, which consistently produced auxins in root exudates (22.5 ± 9.7 µg/mL). Although *R. glutinis* Y23 was a high auxin producer in culture with L-Trp (52 ± 4.2 µg/mL), negative effects on *A. thaliana* plants were observed by inhibiting their development and causing leaf necrosis and yellowing (Figure 3A). A similar phenomenon was found by Fernandez-San Millan et al. [57], who reported that some strains of *Lachancea thermotolerans*, *Meyerozyma guilliermondii*, and *Torulaspora delbruekii* were harmful to *N. benthamiana* seedlings. The reasons for these findings are unclear; however, it is possible that yeasts compete with plants for nutrients or that some fungal-derived metabolites at high concentrations may be phytotoxic under in vitro conditions.

To understand whether the positive effects of yeast were mediated by auxin production, we used the *A. thaliana* DR5:*uidA* line, which is an efficient IAA biosensor. It has TGTCTC motifs commonly found in the promoters of auxin-inducible genes [58]. The roots of *R. glutinis* Y23 were not included because plants showed deficiencies during their development, as explained above. Histochemical analysis showed β-glucuronidase activity in the lateral roots of *A. thaliana* in co-culture with *Solicoccozyma* sp. RY31, *C. lusitaniae* Y11*,* and *Naganishia* sp. Y52 (Figure 4). This supports that the auxins produced by yeasts were responsible for the changes in the root structure.

### 2.5. Evaluation of Auxin-Producing Yeasts as Bioinoculants

Microbes, bacteria, or fungi with in vitro growth-promoting capabilities need to be evaluated on both model plants and crops to fully understand the potential of a strain for use in agriculture. Experiments in pots with vermiculite were carried out to measure the effect of individual auxin-producing yeasts on the corn varieties from which they were grown. After 30 days of treatment with the bioinoculant, a drastic increase in plant development was observed.

In red maize plants, the strains tested had similar values of increased plant height, and their fresh weight was up to 1.5 times greater than the non-yeast controls. However, at the root length level, *Naganishia* sp. Y52 did not induce significant changes, and *C. lusitaniae* Y11 showed the best promotion capacity (Figure 5A). We isolated the Y11 strain from the phyllosphere as well as other strains that have been remarkable for their production of 2-phenylethyl alcohol and 2-phenylethyl acetate that can promote the growth of *Agave*, *A. thaliana* and *N. attenuate* [26].

We observed that all strains of the blue corn plants showed improved plant height and root length, the most significant of which were *N. liquefaciens* Y52 and *R. glutinis* Y23 (Figure 5B). In the field, this would be beneficial to increase the yield of Mexican landraces by developing roots with larger surface areas and improved efficiency in soil nutrients and water uptake. *R. glutinis* Y23 was the only strain that significantly increased fresh plant weight in blue corn. This finding was unexpected, as this yeast was harmful to *A. thaliana*. The genus *Rhodotorula* is well known for its ability to degrade coumaric acid, ferulic acid, vanillic acid, and 4-hydroxybenzoic acid [59], all of which are secreted by maize roots [60]. This may have given *R. glutinis* Y23 an advantage in feeding on exudates in the rhizosphere while also promoting maize growth.

In our work, yeasts were beneficial for both maize landraces, although there were slight variations among strains. This may be due to the ability of each species to colonize plant tissues, produce phytostimulants, or uptake nutrients produced by the roots. The inoculation of *W. saturnus* in maize cv. Merit resulted in increased weight and length of both roots and shoots [50]. Furthermore, *S. cerevisiae* Sc-6 and *Lachancea thermotolerans* Lt-69 improved the height, chlorophyll content, and fresh weight of maize cv. Julliet, as opposed to *Debaryomyces hansenii* Dh-17, which caused harmful effects [57].

Biofertilizers are often Synthetic Microbial Communities (SynCom) made from strains with complementary plant promotion capabilities that ensure beneficial effects on diverse crops [61]. SynCom has significant and positive impacts on maize development [62]. In order to progress this research, it would be interesting to explore the combined effect of our yeast strains to improve corn crop yields in the field.

## 3. Materials and Methods

### 3.1. Maize Accessions

During the 2017–2018 crop cycle, plants of the Mexican maize landrace “Raza Cónico” (Figure 6) were collected from an agricultural field in the locality of “La Paila, Tezoncualpa” in the state of Hidalgo, Mexico (20°31′41.9″ N, 98°43′17.7″ W). This landrace has high emergence vigor and cold tolerance with a predominant distribution in temperate areas between 1800 and 3000 m above sea level [63]. Leaf, root, and grain samples were taken from plants (R4 stage) with no chlorosis, necrosis, or parasite infestation and stored in polystyrene bags at 4 °C until they were analyzed. Twelve plants of each variety were sampled.

### 3.2. Isolation and Identification of Yeast Strains

#### 3.2.1. Yeast Strain Isolation

For phyllosphere yeast communities, 10 g of leaves from each plant were shaken into flasks with 90 mL of sterile distilled water on a rotary shaker (Inch Incubated Shaker, BEING Scientific Inc., Ontario, CA, USA, BIS-3, 230 V) for 30 min; the solution was collected, and serial dilutions were made to 10^−3^ and spread on a YPDA medium (g/L; 10 yeast extract, 20 peptones, 20 D-glucose, 20 agar, 10% tartaric acid (*w*/*v*), pH 4.0). Endophytes were isolated according to the suggestion of [29,64], although slight modifications were made to these. Ten grams of leaves, roots or kernels were surface-sterilized by immersion in 70% ethanol for 30 s and then in 4% sodium hypochlorite solution for 6 min. Plant tissues were then crushed in a sterile blender (Classic Blender Model, Oster, Boca Raton, FL, USA) with 90 mL of sterile distilled water for 5 min, and serial dilutions up to 10^−3^ were spread on a YPDA medium. All plates were incubated at 28 °C for 5 days. Yeast colonies were selected based on morphological differences and purified on a YPDA medium. The purified strains were preserved in microtubes with 30% glycerol at −80 °C.

#### 3.2.2. Molecular Identification and Phylogenetic Analysis

Genomic DNA was isolated from fungal cultures with 5 days of growth on YPDA, using the Quick-DNA Fungal/Bacterial Miniprep Kit (Zymo Research, Irvine, CA, USA), following the manufacturer’s instructions. The region from the Internal transcribed spacer 1 (ITS1) to the D1/D2 domain of large-subunit (LSU) ribosomal DNA (rDNA) was amplified by PCR with the primers ITS1 and NL4 [65,66] in a thermocycler Veriti (ThermoFisher, Waltham, MA, USA). The amplicons were purified and sequenced by Macrogen in Seoul, South Korea. The raw sequences were edited with BIOEDIT 7.2.5 [67] and then clustered into ITS haplotypes using the DAMBE7 package [68]. Taxonomically related sequences were obtained by BLAST in NCBI. All sequences were aligned using the ClustalX program, and the best nucleotide substitution model was determined using JModelTest2 [69]. A phylogenetic analysis was performed in MEGAX with the maximum likelihood method and 1000 bootstrap replicates [70]. All sequences obtained in this study were deposited in the GenBank under accession numbers MN299221–MN299307.

### 3.3. Plant Growth-Promoting Characteristics

#### 3.3.1. Solubilization of Phosphate

Pikovskaya agar plates [g/L; 0.5 (NH_4_)_2_SO_4_, 0.2 KCl, 0.1 MgSO_4_- 7 H_2_O, 0004 MnSO_4_-H_2_O, 0.2 NaCl, 10 D-glucose, 0.002 FeSO_4_ -7H2O), 0.5 yeast extract, 0.5 Ca_3_(PO_4_)_2_, 18 agar] were inoculated with 5 µL of yeast solution (1 × 10^8^ cells/mL). Plates were incubated for 7 days at 28 °C, and the colonies with clear halos surrounding them were considered positive [71]. Three experiments were performed for each isolated strain.

#### 3.3.2. Siderophore Quantification

Siderophore production was induced by inoculating 1 × 10^8^ yeast cells/mL in 5 mL of Iron deferrated Grimm-Allen liquid media [72] and incubating this at 28 °C for 15 days at 120 rpm. The cultures were centrifuged, and 100 µL of the supernatant was mixed with 100 µL of chrome azurol S (CAS)-Fe solution and 2 μL of shuttle solution (0.2 M 5-sulfosalicylic acid) [73]. The reaction mixture was incubated in the dark at room temperature for 10 min and then read at 630 nm on a VersaMax™ Tunable Microplate Reader. The results were reported as percentage siderophore units (% SU) according to the formula suggested by Chowdappa et al. [74]. Quantification was carried out in triplicate.

#### 3.3.3. Production of Plant Cell Wall-Degrading Enzymes

Amylase and pectinase activities were performed in a minimal medium using 2% potato starch (Sigma Aldrich, Burlington, MA, USA) and 1% pectin from the citrus peel (Sigma Aldrich, Burlington, MA, USA) as a carbon source, respectively [75]. Cellulase production was evaluated in Congo red medium with 2% methylcellulose [76] and protease production in skim milk medium [77]. All plates were inoculated with 5 µL of yeast solution (1 × 10^8^ cell/mL) and incubated at 28 °C for 5 days. In order to detect amylases, the colony was flooded in an iodine solution (1% *w*/*v*). The presence of a clear halo around the colony was considered positive. Pectinase activity was visualized by a clear halo around the colony after it was flooded with a CTAB solution (cetyltrimethylammonium bromide, 5% *w*/*v*). Cellulase and protease activities were demonstrated by a clear halo around the colony. The enzymatic index (EI) was determined by subtracting the colony’s diameter from the halo’s diameter [64]. Assays were performed in triplicate for each strain.

### 3.4. Analysis of Auxin Production

#### 3.4.1. Auxin Production in Culture Media

The auxin production was tested in 5 mL of YPD broth (g/L; 10 yeast extract, 20 peptone, 20 dextrose, pH 6.0) with or without 0.1% L-tryptophan (L-Trp) inoculated with 1 × 10^8^ yeast cells/mL. The cultures were incubated for 15 days at 28 °C at 120 rpm. Following this, the supernatants were obtained by centrifugation and filtered through a 0.2 µm filter (Millipore, Burlington, MA, USA). One hundred microliters of the filtered supernatant were mixed with 100 µL of Salkowski reagent (2% of 0.5M FeCl_3_ in 35% HClO_4_) and incubated in the dark at 28 °C for 30 min. The intensity of the pink color formed was read on a VersaMax™ Tunable Microplate Reader at wavelength 530 nm. Following the suggestion of Herrera-Quiterio et al. [77], a commercial IAA (Sigma Aldrich, Burlington, MA, USA) standard curve was made to estimate the auxin concentration produced by each strain. Three experiments for each isolated strain were performed.

#### 3.4.2. Induction of Auxin Production by Root Exudates

In order to obtain maize root exudates, 10 grains of both varieties were disinfected and germinated according to the process adopted by Rosenblueth and Martínez-Romero [78]. Two-day-old seedlings were transferred to a stainless-steel pedestal inside glass tubes (25 × 200 mm) with 50 mL of Fahraeus hydroponic solution (g/L; 0.132 CaCl_2_, 0.12 MgSO_4_·7H_2_O, 0.1 KH_2_PO_4_, 0.075 Na_2_HPO_4_·2H_2_O, 0.005 ferric citrate and 0.07 mg of MnCl_2_·4H_2_O, CuSO_4_·5H_2_O, ZnCl_2_, H_3_BO_3_, and Na_2_MoO_4_·2H_2_O) [79]. Plants were cultivated in a light/dark photoperiod (16 h/8 h) for 5 days at 28 °C. Root exudates were collected using a miraclot filter and subsequently dried by rotary evaporation in an Eppendorf Concentrator 5301 (Eppendorf, Hamburg, Germany) at 60 °C for 1 h and stored at −80 °C. Five mL of YPD broth was supplemented with the root exudates (final concentration approximately 1 mg/mL) and inoculated with 1 × 10^8^ yeast cells/mL. Cultures were incubated for 15 days at 28 °C at 120 rpm, and IAA production was determined from the supernatants as described above. Three experiments for each of the four auxin-producing yeasts were performed. A negative control containing YPD with non-inoculated exudates showed no reaction with the Salkowski reagent.

### 3.5. Arabidopsis Thaliana-Yeast Interaction Assays

Seeds from the *Arabidopsis thaliana* ecotype Columbia-0 (Col-0) and the DR5:*uidA* line were disinfected, according to Rosenblueth and Martinez-Romero [78]. They were then transferred to plates with 0.2× Murashige and Skoog (MS) medium with 0.8% agar and 1% sucrose. Plates were maintained at 21 °C, 16/8 h light/dark photoperiod until germination. Four days later, seedlings were inoculated with 10^6^ cells of the auxin-producing yeasts 4 cm from the root tip. After one week, the length of the primary root and lateral roots emerged, as expected by Matus-Acuña et al. [80]. Thirty plants were evaluated for each yeast and non-inoculated control.

The activation of auxin response in *Arabidopsis* was evaluated after they had interacted with auxin-producing yeasts. For this, plants of the DR5:*uidA* line were washed with Z-buffer (100 mM Na_2_HPO_4_-NaH_2_PO_4_,1 mM MgSO_4_, 10mM KCl, pH 7.4) and then immersed in X-Gluc solution [(100 mM Na_2_HPO_4_-NaH_2_PO_4_ pH 7.0, 0.5 mM K_3_ [Fe (CN)_6_], 0.5 mM K_4_ [Fe (CN)_6_], 10 mM EDTA, 0.5 mM triton X-100, and 1 mg/mL of X-Gluc in DMSO (5-bromo-4-chloro-3-indolyl β-D-glucuronide)] for 16 h at 37 °C in the dark. Plants were washed twice with buffer Z, four times with 70% ethanol, and finally with acetone:methanol (3:1). β-glucuronidase activity, indicative of auxin response, as evidenced in the roots by a blue signal [81]. Ten plants were evaluated for each yeast and non-inoculated control.

### 3.6. Evaluation of Auxin-Producing Yeasts as a Bioinoculant

Seeds from each maize variety were disinfected and germinated, as outlined in previous sections. Three-day-old seedlings were transferred to pots with 1 kg of vermiculite autoclaved (3 times for 30 min at 121 °C, 15 psi) and then individually inoculated with 30 mL of a solution of auxin-producing yeasts (1 × 10^8^ yeast cells/mL). Plants were kept in a growth room at 28 °C with a 16/8-h light/dark photoperiod. They were watered every 3 days with 30 mL of Fahraeus solution [82], which was supplemented with 0.08 g/L NH_4_NO_3_. After 30 days, plants were harvested, and morphometric parameters such as plant height, root length, and fresh weight were measured. The experiment included 5 replicates of the two maize varieties for each of the auxin-producing yeasts alongside two treatments without inoculation as controls. A total of 50 plants were analyzed.

### 3.7. Statistical Analysis

All statistical analyses and graphics were performed in GraphPad Prism 9. Enzymatic activities, plant growth-promoting characteristics, and auxin production were evaluated in three independent experiments, and results were compared between fungal genera using an ANOVA test with Tukey’s post hoc. For bioassays with *A. thaliana* and maize plants, Dunnett’s test was conducted to compare the yeast treatments and the control without inoculation. A value of *p* < 0.05 was considered to be statistically significant.

## 4. Conclusions

This study showed yeast diversity in different maize tissues of two Mexican landraces, with a predominant isolation of *Clavispora lusitaniae*. A high frequency (32%, 12 isolates) of *Rhodotorula glutinis* was observed in red maize and *Papiliotrema flavescens* in blue maize (20%, 10 isolates). *Solicoccozyma* sp. RY31, *C. lusitaniae* Y11, *R. glutinis* Y23, and *Naganishia* sp. Y52 were efficient auxin producers, and when inoculated in *A. thaliana,* they induced the expression of auxin-responsive genes. Inoculation of auxin-producing yeasts in maize plants improved height, fresh weight, and root length, showing that they are suitable as biofertilizers.

## Figures and Tables

**Figure 1 plants-12-01328-f001:**
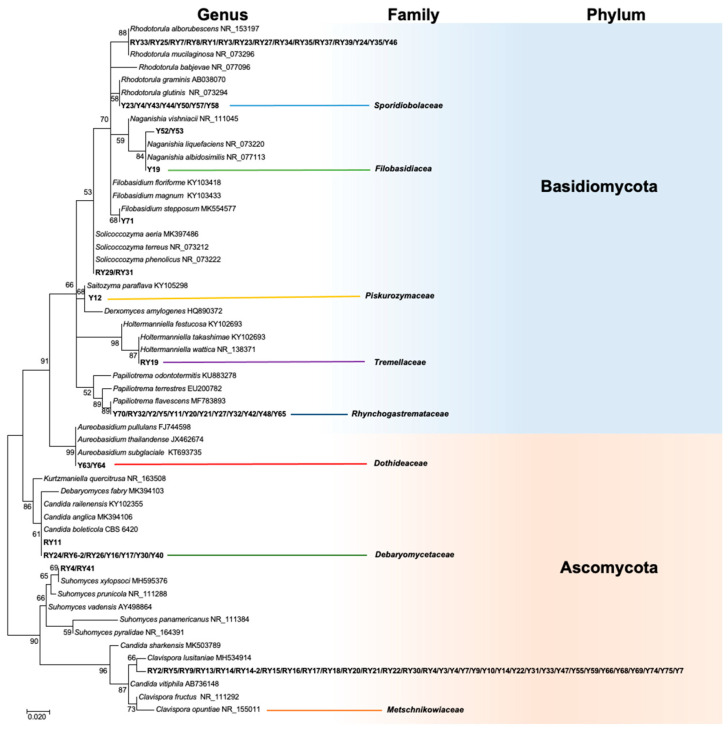
Phylogenetic tree of ITS sequences of yeast isolated from maize based on the maximum likelihood method. The Kimura-2-parameter model was used as the nucleotide substitution model. The numbers at the nodes indicate bootstrap values of 1000 replicates. Branch lengths are proportional to the number of substitutions per site (see scale bar). Strains isolated in this study are in bold.

**Figure 2 plants-12-01328-f002:**
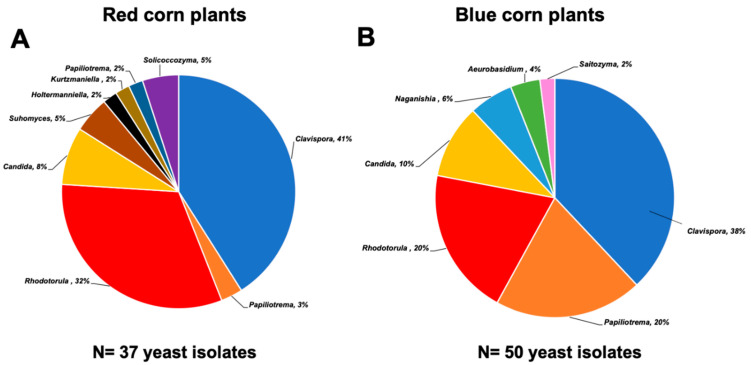
Yeast diversity associated with the Mexican maize landrace “Raza cónico.” (**A**) Relative abundance of yeast genera isolated from plants of red corn (37 isolates) and (**B**) blue corn varieties (50 isolates).

**Figure 3 plants-12-01328-f003:**
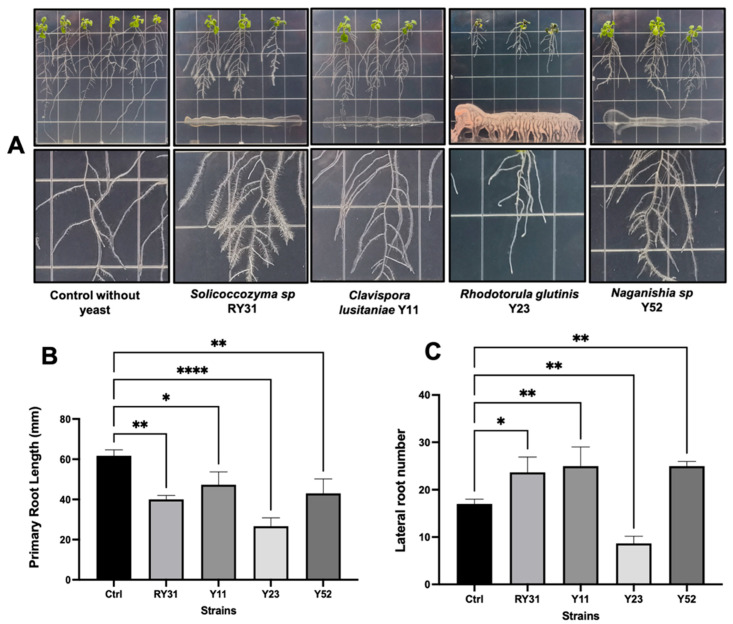
Effect of auxin-producing yeasts on growth and root architecture of *Arabidopsis thaliana* col-0. (**A**) Development of *A. thaliana* seedlings and lateral root production at 7-day post inoculation with yeasts in Murashige and Skoog plates. (**B**) Comparisons of primary root size and (**C**) number of lateral roots between non-inoculated plants and yeast treatments. Statistically significant differences are denoted with asterisks according to Dunnett’s test, with values of *p* < 0.05 (*), *p* < 0.01 (**), and *p* < 0.0001 (****).

**Figure 4 plants-12-01328-f004:**
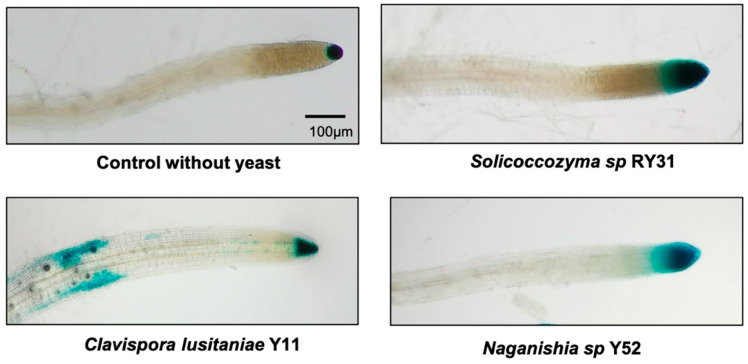
β-glucuronidase activity in roots of *A. thaliana* DR5:*uidA* in the presence of auxin-producing yeast. The blue signal in lateral roots is evidence of activation of the auxin reporter. Roots were collected from seedlings at 7 days post inoculation with yeast and stained with X-Gluc solution.

**Figure 5 plants-12-01328-f005:**
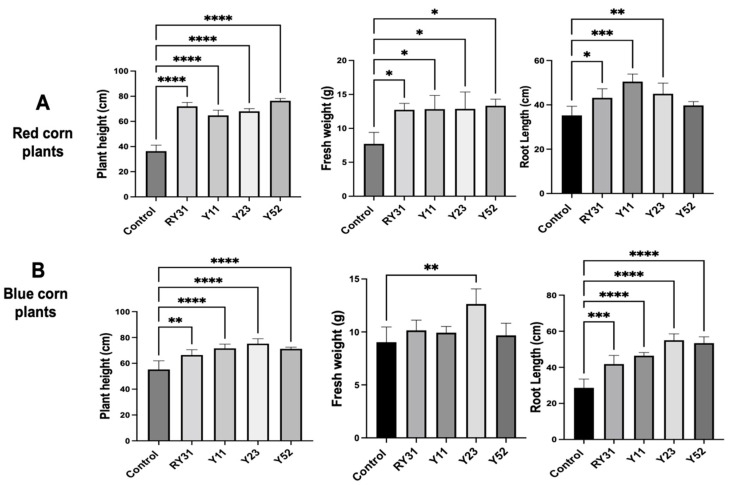
Effect of auxin-producing yeast inoculation on maize growth. (**A**) Morphological parameters of 30-day-old maize plants of red and (**B**) blue varieties. Statistically significant differences are denoted with asterisks according to Dunnett’s test, with values of *p* < 0.05 (*), *p* < 0.01 (**), *p* < 0.001 (***), and *p* < 0.0001 (****). The inoculated strains were *Solicoccozyma* sp. RY31, *Clavispora lusitaniae* Y11, *Rhodotorula* glutinis Y23, and *Naganishia* sp. Y52. Controls were non-inoculated plants.

**Figure 6 plants-12-01328-f006:**
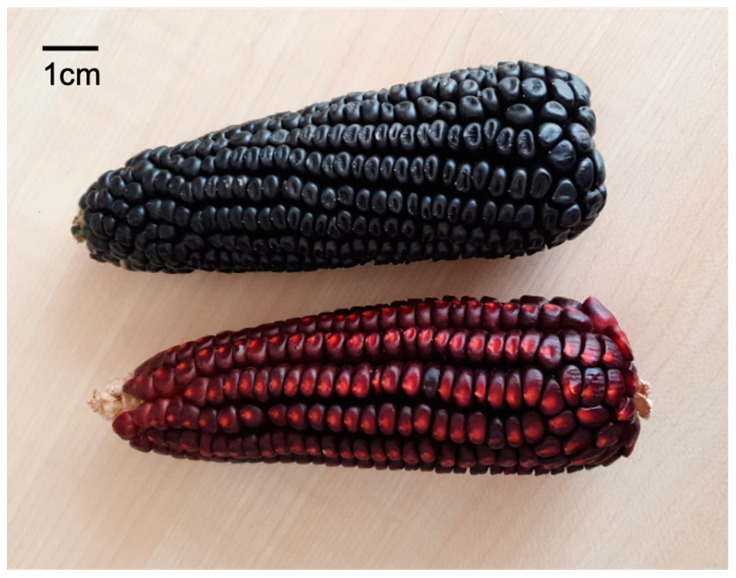
Varieties of the Mexican maize landrace “Raza cónico” were used in this study. Blue (above) and red (below) conical cobs with kernels semi-crystalline and semi-dentate.

**Table 1 plants-12-01328-t001:** Auxin production by yeasts under different culture conditions for 15 days at 28 °C ^‡^.

Strain	YPD Broth (µg/mL)	YPD Broth + L- Tryptophan (µg/mL)	YPD Broth + Root Exudates of Red Maize (µg/mL)	YPD Broth + Root Exudates of Blue Maize (µg/mL)
*Solicoccozyma* sp. RY31	9.3 ± 2.4 b	27.1 ± 1.6 b	22.5 ± 9.7 a	14.1 ± 3.0 b
*Clavispora lusitaniae* Y11	4 ± 0.3 d	11.9 ± 1.9 c	1.3 ± 0.2 c	5.9 ± 0.3 d
*Rhodotorula glutinis* Y23	7.1 ± 0.1 c	52 ± 4.2 a	8.3 ± 0.5 b	9.36 ± 1.2 c
*Naganishia* sp. Y52	15.8 ± 1.8 a	22.3 ± 0.6 b	19.0 ± 7.7 a	20 ± 7.1 a

^‡^ Auxins were quantified in yeast cultures using Salkowski’s reagent and a standard curve with indole-3-acetic acid. The mean ± standard deviation of three independent experiments is shown. Different lower-case letters between strains indicate statistically significant differences (*p*-value < 0.05) according to ANOVA with Tukey post hoc.

## Data Availability

The authors declare that all relevant data supporting the findings of this study are included in this article.

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
