# Peer review of "Mycobiota of Mexican Maize Landraces with Auxin-Producing Yeasts That Improve Plant Growth and Root Development"

_plants, 2023, doi:10.3390/plants12061328_

Round 1

Reviewer 1 Report

The aim of the paper was to use yeast as a plant growth promoting tool to increase maize growth.

This is a very well written paper, and I enjoyed reading it.  This is a very interesting article touching a very hot and applied point which is the use of biological fertilizers. I consider the topic of this paper is original and relevant to the field and suitable for the journal.  The paper addressed a specific gap in the field of organic farming. The conclusions are consistent with the evidence and arguments presented and the authors addressed all the main question posed. Please make all tables self-explanatory and do not use any abbreviation in the table footnote or in the table legend.  Please make all figures self-explanatory and do not use any abbreviation in the figure footnote or in the figure legend.  All references are appropriate, but I suggest adding more 2022 and 2023 references.

Specific Additional Comments

Keywords

Please change the title. The title is not attractive and NOT scientific.

Please arrange all key words in alphabetical order.

Please enlarge figure 1. It is NOT clear and needs to be enlarged.

I’m not sure if the journal accepts results and discussion as ONE unit or not. For me I do not like this system and I recommend that the author write the results in a separate section and also the discussion in a separate section

Please add new references 2023 please.

The authors also missed these important references which is in the field of the work

https://doi.org/10.1007/s10267-005-0268-2

https://doi.org/10.1046/j.1365-2672.2003.02043.x

Please make all tables self-explanatory.

Please make all tables self explanatory.

Please make all figures self-explanatory.

The references section needs critical revision.  The majority of the journal names are NOT capitalized as below. Please correct.

In addition many scientific names in the references must be written in italic format

Additional comments on the tables and figures. All these must be rechecked again and make them all self-explanatory

In order to improve the quality of the paper update the reference list by adding 2022 and 2023 references.

Please follow the instruction to authors in how they write the reference in the list. For references about textbooks, please add the page numbers of the textbook. Also please add the city of the publisher.

Author Response

Reviewer 1

The aim of the paper was to use yeast as a plant growth promoting tool to increase maize growth. This is a very well written paper, and I enjoyed reading it.  This is a very interesting article touching a very hot and applied point which is the use of biological fertilizers. I consider the topic of this paper is original and relevant to the field and suitable for the journal.  The paper addressed a specific gap in the field of organic farming. The conclusions are consistent with the evidence and arguments presented and the authors addressed all the main question posed. Please make all tables self-explanatory and do not use any abbreviation in the table footnote or in the table legend.  Please make all figures self-explanatory and do not use any abbreviation in the figure footnote or in the figure legend.  All references are appropriate, but I suggest adding more 2022 and 2023 references.

Dear reviewer, Thank you very much for your excellent summary. We appreciate each of your suggestions for improving the manuscript. Below is the point-by-point response to each of your comments.

Specific Additional Comments

Keywords

Please change the title. The title is not attractive and NOT scientific.

The title was modified to indicate that it is a study about the mycobiota of landraces corn and at the same time showing the most relevant findings of our work.

Please arrange all key words in alphabetical order.

Done.

Please enlarge figure 1. It is NOT clear and needs to be enlarged.

Done.

I’m not sure if the journal accepts results and discussion as ONE unit or not. For me I do not like this system and I recommend that the author write the results in a separate section and also the discussion in a separate section

Thank you very much for your recommendation.  The journal allows us to link the results with the discussion. We decided that they should be together. However, we have contrasted our data with those obtained in other papers and provided explanations for the observed results. In addition, we included many more papers to enrich the discussion, among them the references you recommended. Thank you very much.

https://doi.org/10.1007/s10267-005-0268-2

https://doi.org/10.1046/j.1365-2672.2003.02043.x

“Discussion: Authors should discuss the results and how they can be interpreted in perspective of previous studies and of the working hypotheses. The findings and their implications should be discussed in the broadest context possible and limitations of the work highlighted. Future research directions may also be mentioned. This section may be combined with Results.” https://www.mdpi.com/journal/plants/instructions

Please add new references 2023 please.

Done

The authors also missed these important references which is in the field of the work

https://doi.org/10.1007/s10267-005-0268-2

https://doi.org/10.1046/j.1365-2672.2003.02043.x

Both were included.  Very interesting work. Thank you very much

Please make all tables self-explanatory.

Done. We include more details in the tables and figures to make them more comprehensible.

The references section needs critical revision.  The majority of the journal names are NOT capitalized as below. Please correct.

References were corrected according to the journal's guidelines.

In addition, many scientific names in the references must be written in italic format

Done

Additional comments on the tables and figures. All these must be rechecked again and make them all self-explanatory

Done

In order to improve the quality of the paper update the reference list by adding 2022 and 2023 references.

Done

Please follow the instruction to authors in how they write the reference in the list. For references about textbooks, please add the page numbers of the textbook. Also please add the city of the publisher.

Done

Reviewer 2 Report

The manuscript entitled “Mexican maize landraces harbor auxin-producing yeasts that improve plant growth and root development” is interesting and relevant to the journal’s scope. However, I feel numerous flaws in different sections of the ms and therefore recommend a through revision/justification is required prior to publication.

In abstract section: Summarize the key results. If possible, add the numerical description of most important results. Be consistent regarding units.

Introduction section contains various unnecessary statements. Remain focused on the topic. Begin with the broadest scope and get progressively narrower, leading steadily to the statement of objectives. Highlight research gap and be clear regarding objectives.

Check the usage of tenses.. In this work, we examined the plant growth-promoting activities of yeasts …….

We also explored the auxin biosynthesis……………..

In results, better to add the numeric description of results (% variations) instead of just adding the data values for easy understanding of the readers.

I suggest separating the Discussion section from Results. Discussion should be merely based on the observed findings. Not just a review of literature. Answer the question posed in introduction and correlate your finding with the existing knowledge.

Material and Method section: Several points need to be clarified. The methodology section lacks the details on experimental design.

Experimental conditions?

Conclusion: Just report the key findings… it should not be a detailed summary of the work done. It seems lengthy.

Check whether the format of all references is according to the journal format.

Language needs substantial improvement. There are several grammatical and typo mistakes throughout the manuscript.

All the tables and figures should be self explanatory. Define all the abbreviations in the table foot note/figure captions.

Author Response

Reviewer 2

The manuscript entitled “Mexican maize landraces harbor auxin-producing yeasts that improve plant growth and root development” is interesting and relevant to the journal’s scope. However, I feel numerous flaws in different sections of the ms and therefore recommend a through revision/justification is required prior to publication. 

Dear reviewer, we appreciate your comments for improving the manuscript. Below is the point-by-point response to each of your comments.

In abstract section: Summarize the key results. If possible, add the numerical description of most important results. Be consistent regarding units.

Thank you very much for the suggestion. The abstract was improved.

Introduction section contains various unnecessary statements. Remain focused on the topic. Begin with the broadest scope and get progressively narrower, leading steadily to the statement of objectives. Highlight research gap and be clear regarding objectives.

Thank you very much for the suggestion The introduction was modified to focus on agriculture, corn, yeasts as bioinoculants and auxin production.

In results, better to add the numeric description of results (% variations) instead of just adding the data values for easy understanding of the readers. 

The results were better explained so that they can be understood by the readers.

I suggest separating the Discussion section from Results. Discussion should be merely based on the observed findings. Not just a review of literature. Answer the question posed in introduction and correlate your finding with the existing knowledge. 

Thank you for your feedback. We decided that the results would be joined with discussion since the journal guidelines allow it, however, following your comments, we contrasted our results with already published papers. We also gave possible explanations for the results obtained, trying to solve the existing gaps in the area of yeast-based biofertilizers.

“Discussion: Authors should discuss the results and how they can be interpreted in perspective of previous studies and of the working hypotheses. The findings and their implications should be discussed in the broadest context possible and limitations of the work highlighted. Future research directions may also be mentioned. This section may be combined with Results.” https://www.mdpi.com/journal/plants/instructions

Material and Method section: Several points need to be clarified. The methodology section lacks the details on experimental design. Experimental conditions? 

The methodology was more detailed and included the repetitions per experiment and the conditions under which they were performed.

Conclusion: Just report the key findings… it should not be a detailed summary of the work done. It seems lengthy. 

Done. The conclusions were summarized by highlighting the most important findings.

Check whether the format of all references is according to the journal format. 

Done, the references were modified according to the journal format

Language needs substantial improvement. There are several grammatical and typo mistakes throughout the manuscript

The English was reviewed by a native speaker and errors were corrected.

All the tables and figures should be self-explanatory. Define all the abbreviations in the table foot note/figure captions.

Done. Added more details in the tables and figures to make them more understandable.

Reviewer 3 Report

Compared to agrochemicals, bioinoculants based on plant microbiomes are a sustainable option for increasing crop yields and soil fertility. From the Mexican maize landrace "Conico", the authors identified yeasts and evaluated in vitro their ability to promote plant growth. Inoculation 17 tests were performed on maize and morphological parameters were measured in this study. Their results showed that the inoculation of auxin-producing yeasts led to an increase in maize plant height, fresh weight, and root length. Overall, maize landraces harbor plant growth promoting yeasts and have the potential for use as agricultural biofertilizers.

In general, the manuscript is well-written and results are quite significant, I have the only following minor comments:

1.      why author measure the auxin production, not the other plant hormone, what is the reason for it;

2.      literature need to be update, there is no much references from 2022-2023;

3.      the reference format is not consistent, some have DOI number, some not, it should be all consistent.

Author Response

Reviewer 3

In general, the manuscript is well-written and results are quite significant, I have the only following minor comments:

  1. why author measure the auxin production, not the other plant hormone, what is the reason for it;

The importance of auxin production and its role in plant development was further detailed in the introduction and discussion.

But basically, auxins were analyzed because…

“Auxins control several functions in plants, such as vascular tissue differentiation, root initiation, cell division and elongation, hence it is crucial that bioinoculants include strains that produce this phytohormone”

  1. literature need to be update, there is no much references from 2022-2023;

Done. References were updated.

  1. the reference format is not consistent, some have DOI number, some not, it should be all consistent.

References were corrected according to the journal's guidelines.

Round 2

Reviewer 1 Report

The paper can be accepted now. The authors answered all my comments. Well Done.